# Detection of Histamine Dihydrochloride at Low Concentrations Using Raman Spectroscopy Enhanced by Gold Nanostars Colloids

**DOI:** 10.3390/nano9020211

**Published:** 2019-02-06

**Authors:** Eleazar Samuel Kolosovas-Machuca, Alexander Cuadrado, Hiram Joazet Ojeda-Galván, Luis Carlos Ortiz-Dosal, Aida Catalina Hernández-Arteaga, Maria del Carmen Rodríguez-Aranda, Hugo Ricardo Navarro-Contreras, Javier Alda, Francisco Javier González

**Affiliations:** 1Coordinación para la Innovación y Aplicación de la Ciencia y la Tecnología, Universidad Autónoma de San Luis Potosí, 78210 San Luis Potosí, Mexico; samuel.kolosovas@uaslp.mx (E.S.K.-M.); a.cuadrado@pdi.ucm.es (A.C.); joazet.ojeda@uaslp.mx (H.J.O.-G.); aida.arteaga@uaslp.mx (A.C.H.-A.); carmen.rgz.aranda@gmail.com (M.d.C.R.-A.); hnavarro@uaslp.mx (H.R.N.-C.); javier.gonzalez@uaslp.mx (F.J.G.); 2Applied Optics Complutense Group, Faculty of Optics and Optometry, University Complutense of Madrid, Av. Arcos de Jalon, 118, 28037 Madrid, Spain; 3Instituto de Física Luis Terrazas, Benemerita Universidad Autónoma de Puebla, Av. San Claudio, 18, 72570 Puebla, Mexico; 4Doctorado Institucional en Ingeniería y Ciencias de Materiales, Universidad Autónoma de San Luis Potosí, 78210 San Luis Potosí, Mexico; ortiz.dosal.lc@gmail.com

**Keywords:** SERS, histamine, nanostars, nanophotonics, computational electromagnetism

## Abstract

In this paper, we report a fast and easy method to detect histamine dihydrochloride using gold nanostars in colloidal aqueous solution as a highly active SERS platform with potential applications in biomedicine and food science. This colloid was characterized with SEM and UV–Vis spectroscopy. Also, numerical calculations were performed to estimate the plasmonic resonance and electric field amplification of the gold nanoparticles to compare the difference between nanospheres and nanostars. Finally, aqueous solutions of histamine dihydrochloride were prepared in a wide range of concentrations and the colloid was added to carry out SERS. We found SERS amplified the Raman signal of histamine by an enhancement factor of $1.0\times 10^{7}$, demonstrating the capability of the method to detect low concentrations of this amine molecule.

## 1. Introduction

Surface-enhanced Raman spectroscopy (SERS) is a useful technique for the characterization of small groups of molecules near or bound to plasmonic surfaces. It is powerful, non-destructive, and provides information about the chemical structure and identity of materials [1,2,3,4,5]. These capabilities make possible the wide use of SERS in biosensors for the detection of substances of biological interest and pathogens [5,6,7,8,9,10,11], being gold and silver two of the metals that offer better results for this kind of applications [12,13,14,15]. The above-mentioned metals in the form of nanoparticles have the advantage that can be used directly, as colloidal solutions, acting as tridimensional plasmonic systems with customized resonances that can be tuned with the size and shape of the dispersed nanoparticles [4,16,17]. Au nanostars have been proven useful for SERS, they also present unique optical and electric properties. Previous groups have reported the synthesis of Au nanostar with a good degree of symmetry control by using a robust solution-phase method [18], or by increasing the seed concentration in the growth solution [19]. Also, the potential of these nanostructures has been evaluated in biomedical applications by functionalizing its surface using a biocompatible polymer [20]. Among many existing anisotropic gold nanostructures star-shaped nanoparticles (gold nanostars) have achieved a huge interest, mainly due their high biocompatibility, chemical stability and unique optical properties, which makes them useful in a wide range of applications in fields such as plasmonics, spectroscopy, biological applications (bioimaging, biosensing, drug delivery), and catalysis [21,22].

Histamine is a relevant biological substance in medicine and food science. It is a biogenic amine that transmits signals from cell to cell in the skin, intestines, and organs of the immune system. Structural differences in the receptor cell membranes are responsible for different responses to histamine among individuals [23,24].

For example, the interaction between histamine and H1 receptors causes a drop in the blood pressure and muscle contractions, and the interaction with H2 receptors is associated with acidic stomach secretions [25,26]. The consumption of food with a high concentration of histamine may result in intoxication with symptoms such as nausea, diarrhea, headache, asthma, angioedema, urticaria, and itch. These reactions are part of the histamine poisoning, or scombroid poisoning [27,28,29,30]. The concentration of histamine in food should be less than 10 mg/100 g, while the average concentration in human plasma is around 7.2 nmol/L [27,31]. Hence, the determination of histamine concentration that may be present in certain food products is a safety issue of public concern [32].

Conventional methods for determining the presence and concentration of histamine are high-performance liquid chromatography (HPLC), fluorometry, and detection by enzymes (e.g., enzyme linked immunosorbent assay, ELISA) [31]. HPLC and fluorometry involve slow protocols for derivatizing with o-phtalaldehyde or dansyl chloride. Another drawback of the fluorometric assays is that they require methanol extraction and purification with an anion exchange column as a pretreatment. Furthermore, due to the similarity between the structures of histamine and histidine, the measurement tends to have a low selectivity for histamine [8], even when separated previously by HPLC. In contrast, enzyme-based methods provide rapid detection but require the use of unstable enzymes and very expensive test kits, also they may overestimate the amount of histamine [8].

The detection of histamine in food using SERS techniques has been proved recently and opens the way to improved and more reliable detection techniques [33,34]. Our study describes an alternative approach using gold nanoparticles in colloidal suspension which strongly amplifies the Raman signal of extermely low concentrations of histamine dihydrochloride, which corresponds with the dication of the molecule. Previous results using gold and silver nanoparticles for the detection of biological samples have demonstrated the capabilities of the method when using colloids of nanoprisms [11] or surface bounded nanostars [35]. However, the use of colloids with gold nanostars has not been reported so far. In this contribution, results show that the SERS signal of histamine dihydrochloride can be obtained in a fast and accurate way in an aqueous solution, so the results obtained here contribute to the establishment of a useful and valid method for the detection of this biochemical compound.

## 2. Methods

### 2.1. Synthesis of Nanoparticles

The first step to fabricate the gold nanostar colloid is to synthesize gold nanospheres with the Turkevich method [36] and subsequently a second reduction with pH control using a Good’s buffer and hidroxilammonium chloride respectively. An aqueous solution of chloroauric acid 2.5 mM is heated to 95 ${}^{{}^{\circ}}$C, and an aqueous solution of trisodium citrate 2.5 mM (Na${}_{3}$C${}_{6}$H${}_{5}$O${}_{7}$) is added as a reducing agent. This produces a red solution which indicates the formation of gold nanospheres. The second stage of the synthesis consists in growing spikes on the surface of the nanoparticles. The nanospheres solution is added to a 50 mM aqueous solution of 2-[4-(2-hydroxyethyl)piperazin-1-yl] ethanesulfonic acid (HEPES), to control and maintain the pH at a physiological value, and hence the morphology of the nanostructure, and with a 0.1 M aqueous solution of hydroxylammonium chloride (HONH${}_{3}$Cl) as a reducing agent. Finally, the solution is washed with deionized water. The colloid changes from red to blue, signaling the formation of gold nanostars. As Figure 1 shows, generated peaks are distributed on the surface of the nanospheres, generating nanostars measuring around 160 nm in diameter.

### 2.2. Numerical Simulation

In this work, we have performed numerical simulations of the plasmonic optical response using COMSOL Multiphysics. To simplify the calculations, nanostars with a diameter of 150 nm were considered, the peaks were homogenously distributed on the surface of the sphere as conic protuberances, with a height of 25 nm and a maximum diameter of 31 nm, in accordance with the mean values obtained by the SEM images. For an improved study of the optical response of the nanostar, the initial nanospheres 75 nm in radius have been simulated. We evaluate of the optical response through the extinction efficiency, $Q_{\mathrm{ext}}$, which is calculated as:(1)$$Q_{\mathrm{ext}}=\frac{\int_{V}\vec{J}\vec{E}dv+\int_{S}\vec{n}\vec{S}ds}{I_{0}\pi D^{2}/4},$$

The first part of the numerator is the power loss related with the Joule effect, where $\vec{J}$ and $\vec{E}$ are the induced current density and the electric field along the structure, respectively; and the integration is evaluated within the nanostructure. The second part of the numerator is related to the scattered power, where $\vec{n}$ is the normal vector pointing outwards and $\vec{S}$ is the Poynting vector. In this case, the integration surface is a sphere located in the far field region. Finally, $I_{0}$ and *D* are the incident irradiance and the diameter of the particle, respectively.

### 2.3. Spectroscopic Measurements

To complete this analysis, the optical responses of both types of nanoparticles were measured through UV–Vis spectroscopy to compare them with the calculated extinction coefficients. We used a Ocean Optics spectrometer (model USB650) (Largo, FL, USA). To perform SERS, histamine dihydrochloride serial 1:10 successive dilutions were prepared from an initial 3 M stock solution until a $3\times 10^{-7}$ M one was obtained. From these solutions of histamine, new dilutions were made 1:3 with the nanostars colloid, mixed with deionized water, to give the final mixture of histamine and nanoparticles from 1 M to $1\times 10^{-7}$ M. The pH range was maintained in the range of 7.1 to 7.9. The Raman measurements were performed on a Horiba Jobin Ybon XplorRA ONE Raman spectrometer (Irvine, CA, USA) coupled to an Olympus BX41 optical microscope (Ciudad de México, México), using a near-infrared (λ=785 nm) Raman laser source with an average power of 20 mW at the sample location. This laser line has the advantage that strongly minimizes the huge fluorescence background typical of biological samples.

## 3. Results

The synthesis of gold nanostars was successful and 160-nm-diameter structures showing triangular peaks were obtained from gold nanospheres. Figure 1a shows a SEM image of one of the fabricated nanostars. When prepared in colloid form, nanostars coalesce in clusters of several units (see Figure 1b). This is of importance from a spectroscopic point of view because these clusters may distort the maximum peaks of the spectral response.

Through numerical simulations, the spectral response of the nanoparticles used in this work was calculated. The fabrication of gold nanostars uses gold nanospheres as a precursor nanostructure, the plasmonic response of the synthesized nanostructures varied either by increasing the radius of the spheres or by generating small random peaks on its surface. As it happens with the synthesized gold nanoparticles, the spherical geometry is the starting point of the numerical simulations. The nanostar geometry is obtained after adding spikes over the nanosphere surface to resemble the actual shape of gold nanostars.

The numerically calculated optical response of the studied nanostructures is shown in Figure 1c and expressed in terms of the extinction coefficient of the structure. According to Mie theory, gold nanospheres (blue line) show a plasmonic resonance located at 530 nm. This response is red-shifted towards 550 nm for the nanostar, but now the importance of this resonance is smaller than the main peak at 675 nm related with the presence of the protuberances. This main response depends on the height and maximum diameter values of the peaks of the nanostars. To complete this analysis, the optical responses of both types of nanoparticles were measured through UV–Vis spectroscopy to compare them with the calculated extinction coefficients. The absorption spectra are given in Figure 1d. The nanosphere colloid (blue line) presents the expected plasmonic resonance at 530 nm, according to the simulation shown in Figure 1c. As predicted by computational electromagnetism, the optical response of the nanostar (red line) has a resonance centered at 675 nm, and shows broader bandwidth. This effect can be explained considering the morphology variation of the generated nanostars, as well as their size distribution in the sample, and the presence of clusters (see Figure 1b). It is shown that the nanostars have a resonance closer to the wavelength of the incident laser (785 nm), hence, the amplification will be larger for nanostars.

The signal obtained from Raman spectroscopy is proportional to the fourth power of the modulus of the electric field. Therefore, moderate field enhancements, although spatially confined, provide large amplification factors of the Raman response. Figure 2 shows the spectral field enhancement factor for nanostars. We may see that the maximum of it appears at a wavelength $\lambda_{\max}=675$ nm. Our experimental setup excites the Raman spectra using an excitation laser operating at $\lambda_{\mathrm{exc}}=780$ nm, where the field enhancement is about ×80, that is good enough to generate a Raman signal amplification of around $4\times 10^{7}$. Through computational electromagnetism we have evaluated the near field distribution around nanostars when the incidence is having an amplitude of 1 V/m. Therefore, the obtained electric field also represents the field enhancement map of the structure. The results for $\lambda_{\max}$, and $\lambda_{\mathrm{exc}}$ are shown in Figure 2b and Figure 2c respectively. As expected, the electric field is located near the tip of the peaks, achieving a strong enhancement. For comparison, we have evaluated the electric field map generated by a nanosphere (see Figure 2d). In this case, the electric near field has a dipolar distribution, resulting in a field enhancement factor of ×7.4. These results express the goodness of nanostars with respect to nanospheres, because higher field enhancement will strongly increases the capability to detect a Raman shift.

Using the model described in the supplementary material in [37] we calculated a SERS enhancement factor (EF) of $1.0\times 10^{7}$ for the band at 1260 cm${}^{-1}$. This result is somewhat larger than those obtained for a silver film over nanosphere (AgFON) from Wen-Chi Lin et al. ($4.3\times 10^{6}$) [38] and far larger than that obtained from molecularly imprinted polymers by Fang Gao et al. ($1.0\times 10^{4}$) [27]. This also applies even for the values reported in refs. [39,40], being our EF value closer to $1.91\times 10^{7}$, recently reported [41]. As the spectral positions of the bands in Raman and SERS measurements are nearly unchanged, it is reasonable to conclude that the EF produced by the nanostructure is predominantly responsible for the high SERS signal intensity.

Raman spectrum of powder histamine dihydrochloride is shown in Figure 3a as a reference. The Raman shift spectra of most of the vibrational modules of the histamine in solid state and the histamine dihydrochloride coincide. The Raman peaks are compared with Raman vibrational modules of histamine of previous works (Table 1) [38,42,43], and the vibrational modules are found at the same Raman shifts, detecting peaks related to histamine.

Once the nanostars were added to the different solutions of histamine dihydrochloride an increase in the intensity was observed allowing the identification of the characteristic Raman peaks of the histamine molecule that correspond to the vibrational modes of the imidazole ring. Figure 3b and Table 1 allows the identification of these vibrations at low concentrations. For example, the 846 cm${}^{-1}$ corresponds to a bending in the plane of the imidazole ring or in the side chain of the molecule (ring A, wagging C), the peak at 985 cm${}^{-1}$ corresponds to a flexion of the plane of the imidazole ring (ν(N1-H), ν(C2-H)), some other peaks around 1260 cm${}^{-1}$ correspond to flexions in the plane of the NH, and bending in the plane of CH respectively: Sy(N3-H), Sy(C4-H), ν(C2-H), and ν(N1-H) [38,42,43]. These results show how, even at very low concentrations ($10^{-7}$ M), it is still possible to identify histamine thanks to the SERS technique using gold nanostar colloids (see top spectrum at Figure 4).

After performing SERS on different solutions of histamine dihydrochloride, it is observed that the addition of the gold nanostructures increases the intensity of the Raman spectrum (see Figure 3b). This is due to the surface plasmon resonance at the nanostars that produces larger enhancement of the electric fields that subsequently amplify the emitted light from irregular points (gaps, sharp edges, etc.) [1]. Figure 4 shows the Raman spectra for several concentrations of histamine, from 1 M to 10${}^{-7}$ M. A lower concentration means weaker signal. However, the characteristic peaks of the histamine spectrum are revealed allowing its identification. To prove this result, we have made an analysis of the resonance around $\kappa =1265\;{\mathrm{cm}}^{-1}$ (see Figure 5). This Raman shift is representative of the histamine spectra. Around this wavenumber, the phonon modes in the collected Raman spectra were decomposed into Lorentzian lineshapes. Raman spectra were analyzed by first removing the background contribution using a linear function, then, the Lorentzian curves profiles have been used for the fitting procedure of the experimental Raman spectra. This fitting applies a Levenberg–Marquardt least-squares-based iterative algorithm to optimize the parameters of the Lorentzian functions (center, height, and FWHM). These obtained spectra and Lorentzian components are shown in Figure 5a, and the importance of the Lorentzian component centered at κ=1265 cm${}^{-1}$ is presented in Figure 5 as a function of the molarity of the sample. We may see how the Lorentzian at κ=1265 cm${}^{-1}$ remains relevant at every concentration, revealing the presence of histamine in the sample.

## 4. Conclusions

As result of the comparison between the extinction coefficient evaluated from simulation and the measured optical response of nanoparticles, we positively demonstrated the capability of numerical simulations to deal with nanostructures intended to interact with molecules. Because of their reliability, these tools can help speed-up the selection of geometries of nanoparticles in biochemical applications.

The detection and identification of histamine being a serious concern in safety regulations of the food industry, and some other areas of biomedicine, requires the availability of reliable analytical tools. In this contribution, we have demonstrated that Raman spectroscopy amplified by surface plasmon resonance allows qualitative analysis of histamine at extremely low concentrations (as low as 10${}^{-7}$ M) in aqueous dilutions. The resonance takes place at the field-enhancement location of gold nanoparticules having a nanostars geometry and prepared as a colloid. This colloid was added to solutions of histamine dihydrochloride at various concentrations. A simulation was performed with a finite element method to calculate the spectral extinction coefficient, and the field-enhancement due to the nanoparticles. Spectroscopic measurements have proved that spectral absorption fits well with the numerically obtained results supporting the reliability of the simulations. According with the computational evaluation of the field enhancement, an increase was observed in the spectral intensity, as well as an amplification of the characteristic Raman peaks of the histamine molecule. The SERS enhancement factor is $1\times 10^{7}$, which indicates the potential application of the proposed method, allowing lower limits of detection of the molecule.

In summary, this work proposes a reliable method for the detection of histamine in concentrations as low as $10^{-7}$ M, which is a concentration suitable for food quality control applications (concentrations of $\approx 10^{-4}$ M). Further work and refinement of the technique are needed to detect histamine at even lower physiological concentration ($\approx 10^{-9}$ M) using the above-mentioned method.

## Figures and Tables

**Figure 1 nanomaterials-09-00211-f001:**
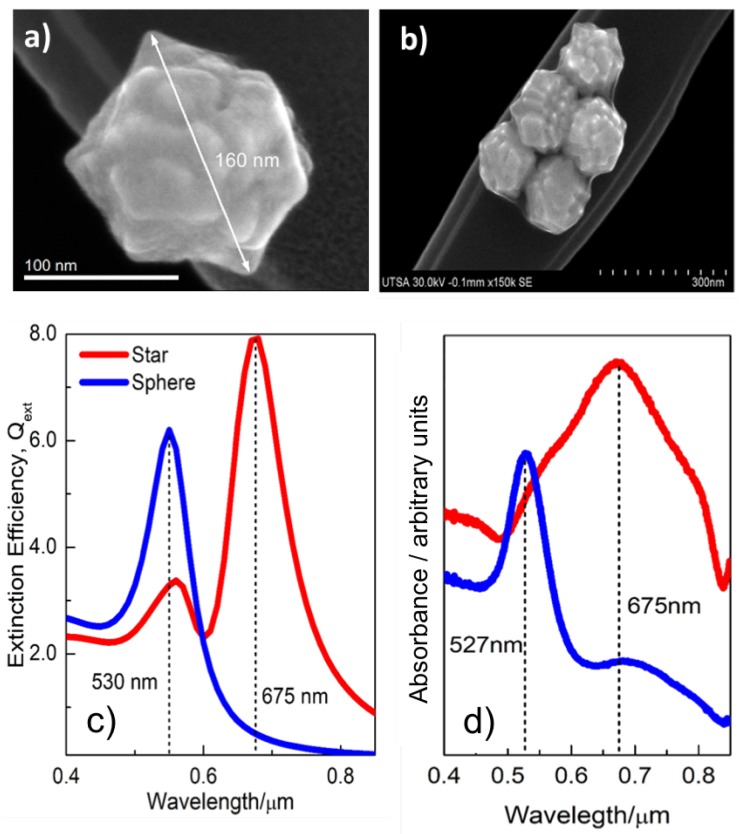
(**a**) Scanning electron micrograph of a gold nanostar with a diameter of about 160 nm. The diameter distribution shows a median value of 159 nm with a standard deviation of 3 nm. (**b**) Clustering of gold nanostar that will be present in the colloid. (**c**) Extinction coefficient evaluated through numerical simulations of both nanospheres (blue) and nanostars (red). (**d**) Measured spectral absorbance of colloids of nanospheres (blue) and nanostars (red) at a concentration of 2.5 mM.

**Figure 2 nanomaterials-09-00211-f002:**
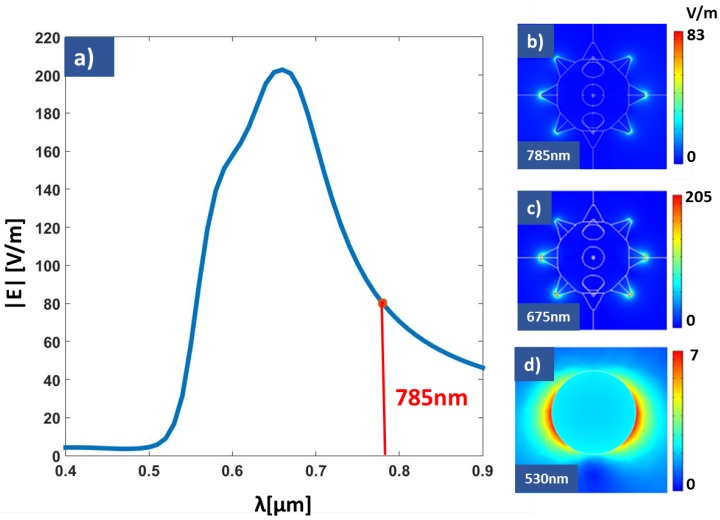
(**a**) Spectral field enhancement for the nanostar geometry. Our fabricated structures shows its maximum of the field enhancement($\lambda_{\max}=675$ nm) slightly shifted from the wavelength of the excitation source of our Raman spectrometer ($\lambda_{\mathrm{exc}}=785$ nm). (**b**–**d**) Near field maps for the nanostar and nanosphere geometries at different wavelengths. (**b**,**d**) are evaluated at the maximum response wavelength for each geometry, and (**c**) is for the excitation wavelength. As far as the input plane wave is having an electric field amplitude of 1 V/m, the near field map also represents the field enhancement.

**Figure 3 nanomaterials-09-00211-f003:**
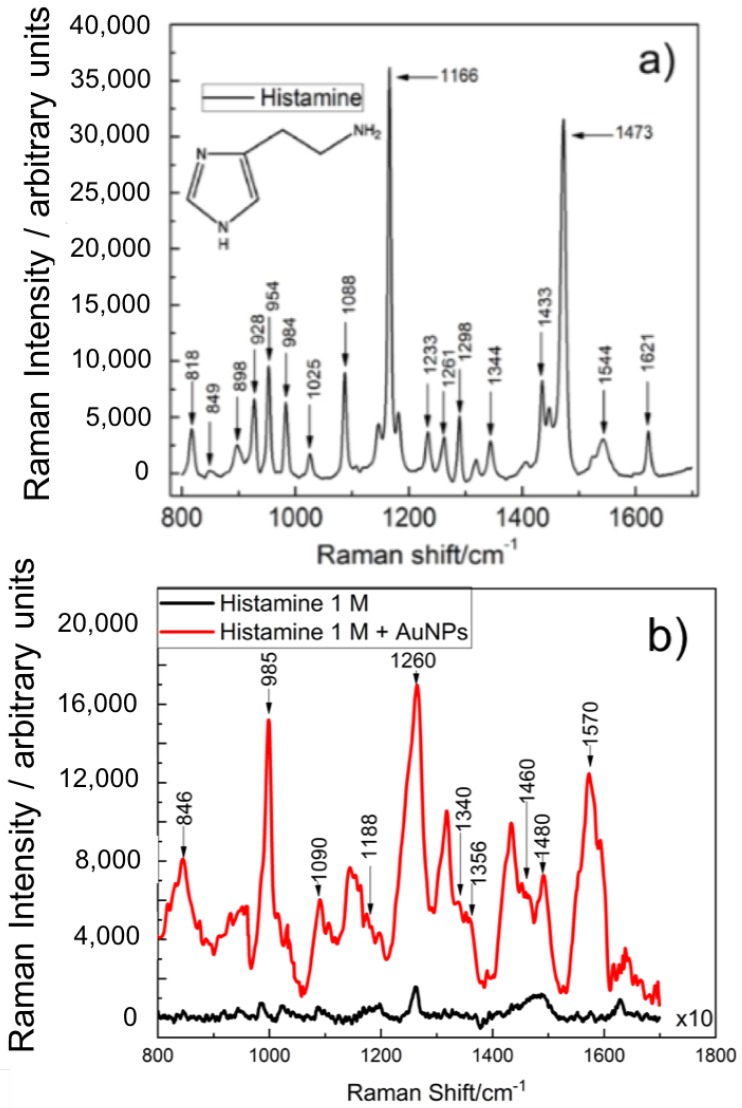
(**a**) Raman spectrum of powder histamine after baseline correction using Vancouver algorithm, where the modes observed correspond with the work reported by Collado et al. [42]. (**b**) Raman spectra of histamine 1 M with (red) and without AuNPs (black).

**Figure 4 nanomaterials-09-00211-f004:**
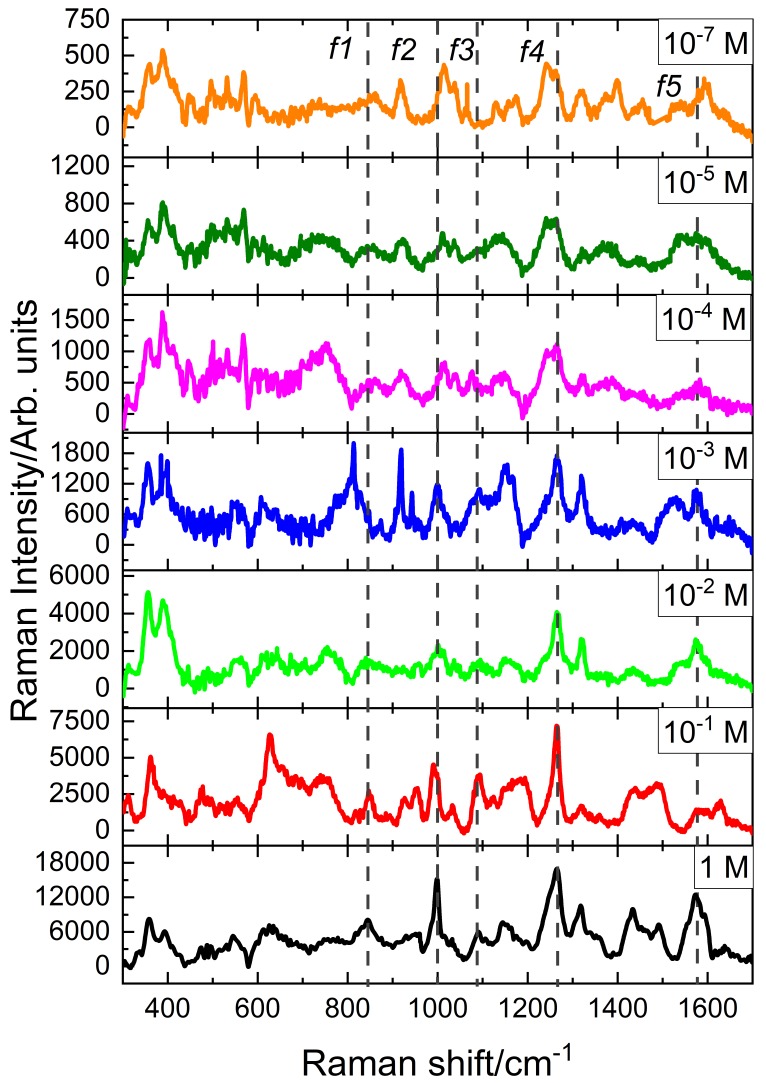
Raman spectra of the vibrational resonances of the histamine dication molecule at several concentrations. All the solutions include gold nanoparticles. Dashed vertical lines corresponde with characteristic peaks of the histamine molecule. Figure 5 will focus our attention on the f4 resonance around $\kappa =1265\;{\mathrm{cm}}^{-1}$.

**Figure 5 nanomaterials-09-00211-f005:**
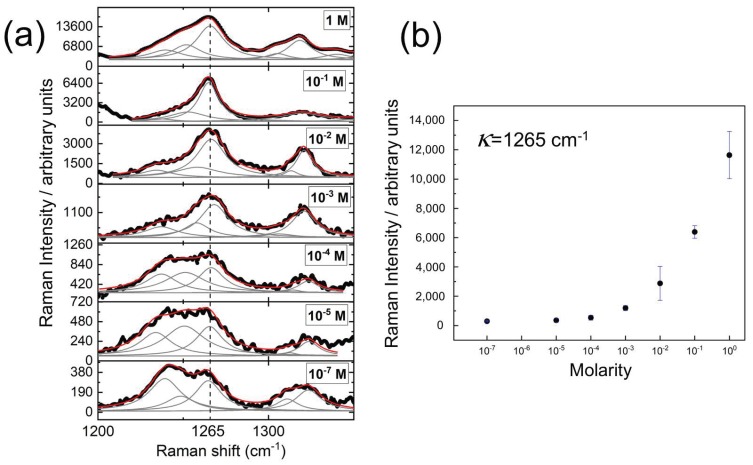
(**a**) Detail of the Raman spectra around $\kappa =1265\;{\mathrm{cm}}^{-1}$ (represented as a dashed vertical line). The gray lines under the curve correspond with the Lorentzian lineshapes obtanied after fitting. The red line represents the results obtained from this decomposition. This fitting has been done for several molarity values from 1 M to 10${}^{-7}$ M. The case of $M=10^{-6}$ has not been prepared nor measured. (**b**) Relative intensity of the $\kappa =1265\;{\mathrm{cm}}^{-1}$ Lorentzian peak as a function of molarity. The error bars represent the variability between samples.

**Table 1 nanomaterials-09-00211-t001:** Comparison in wavenumber, κ (cm−1) of our measurements (first column) with previously published works: Lin et al. [38] (second column), and Torreggiani et al. [43]. Mode notation: ν = stretching; sy = symmetric; τ = twisting

κ (cm${}^{-1}$)	κ (cm${}^{-1}$) [38]	κ (cm${}^{-1}$) [43]	Assignment
846			ring A, wagging C
895			ν (N3-H)
950			ring A
985		980	ν (N1-H), ν (c2-H)
1028	1024.73	1005	sy (C5-H), τ (C2-H), ν (C1-H)
1090	1084.05	1088	sy (N3-H), sy (C5-H)
1188			wagging C, ν (C2-H)
1260		1270	ring breathing
1340			τ (C1-H), ν (N1-H)
1356	1358.07		translation C, ν (C1-H), ν (N1-H)
1438		1435	
1460	1455.46		ν (C4-H), sy (N3-H)
1480			sy (C5-H), sy (N3-H)
1570	1567.61	1579	ν (N3-H)
1629		1618

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
