# Peer review of "Detection of Histamine Dihydrochloride at Low Concentrations Using Raman Spectroscopy Enhanced by Gold Nanostars Colloids"

_nanomaterials, 2019, doi:10.3390/nano9020211_

Round 1
Reviewer 1 Report
It is interesting paper, I will recommend it for publication if the authors clearly address the following points:
- According to Fig. 2, the common 632.8 nm excitation of He-Ne laser will be more suitable for this study than 785 nm? Did the authors try/can try 632.8 nm excitation?
- SERS on colloidal nanoparticles/nanostars in suspension is known to be poorly reproducible. Can the authors comment about reproducibility of their SERS experiments (SERS spectra in Fig. 4)?
- Please provide more details how enhancement factor was calculated. How the number of adsorbed molecules (SERS experiment) have been determined?
- Assignment of the important band at 1260 cm-1 is missing in the Table 1.
- The SERS spectra on Figs 3-5 seems to be baseline corrected? If it is a case, please note it in the figure captions.
- please explain what you mean by "analytical amplification factor 10" at conclusion?
- correct some typos such as "nanostart" at conclusion.
Author Response
Reviewer #1:
It is interesting paper, I will recommend it for publication if the authors clearly address the following points:
OBSERVATIONS
O1. According to Fig. 2, the common 632.8 nm excitation of He-Ne laser will be more suitable for this study than 785 nm? Did the authors try/can try 632.8 nm excitation?
Our Raman Spectrometer HORIBA Xplora Plus includes two laser sources of wavelengths 785 and 532 nm. The more appropriate laser line 632.8 nm is not available in our system. However, having the gold nanostar resonance curve in mind (Fig. 1) we choose the 785 nm laser, because this wavelength is included in the broad resonance response, in opposition to the 532 nm line, which is outside of the main resonance region, with the added benefit that this infrared line strongly minimizes the huge fluorescence background produced by the histamine-AuNP system (or almost any biological samples), which may be very inconvenient and hard to subtract in a reliable way.
Taking in consideration the reviewer’s comments, we decided to implement the following change in the section 2.3. Spectroscopic measurements:
The original paragraph
The Raman measurements were performed on a Horiba Jobin Ybon XplorRA ONE Raman spectrometer coupled to an Olympus BX41 optical microscope, using a near-infrared (λ= 785 nm) Raman laser source with an average power of 20mW at the sample location.
Was changed to:
The Raman measurements were performed on a Horiba Jobin Ybon XplorRA ONE Raman spectrometer coupled to an Olympus BX41 optical microscope, using a near-infrared (λ= 785 nm) Raman laser source with an average power of 20mW at the sample location. This laser line has the advantage that strongly minimizes the huge fluorescence background typical of biological samples.
O2. SERS on colloidal nanoparticles/nanostars in suspension is known to be poorly reproducible. Can the authors comment about reproducibility of their SERS experiments (SERS spectra in Fig. 4)?
With the gold nanostars, we were able to reproduce three times the SERS results, with either the 785 nm, or the green source when controlling for the strong fluorescence, obtaining almost identical results in amplification.
We would like to refer the reviewer to the publication by several of the coauthors of the present manuscript “Diagnosis of Breast Cancer Using Surface-Enhanced Raman Spectroscopy to detect Sialic Acid Concentrations in Human Saliva. A. Hernández-Arteaga, J. J. Zermeño Nava, E. S. Kolosovas-Machuca, J.J. Velázquez-Salazar, E. Vinogradova. M. José-Yacamán and H. R. Navarro-Contreras. NANO RESEARCH, Vol 10(11), November (2017) pp 3662-3670. DOI 10.1007/s12274-017-1576-5,” where we evaluated the reproducibility of the calibration of the amplification of Sialic Acid by Ag-NP, in the supplementary information, where excellent reproducibility was always observed.
O3. Please provide more details how enhancement factor was calculated. How the number of adsorbed molecules (SERS experiment) have been determined?
The theoretical enhancement factor has been obtained through the evaluation of the electric field distribution when illuminating with a plane wave having an amplitude of 1 V/m. Therefore, the value of the field amplitude replicates the field enhancement factor. Then, as far as the Raman response is proportional to the fourth power of the electric field, we calculate this power to obtain the simulated enhancement factor.
The experimental enhancement factor was calculated as follows: a) taking the direct relative spectral factor between the Raman signal produced by the histamine powder, and that of the 0.1 M spectra; b) multiplying by the relative factor in number of histamine molecules illuminated by the laser, contained in the volumes of the diluted colloidal system and that of the solid powder. This approach assumes that all histamine molecules are adsorbed by the gold nanostars. As this number of histamine molecules adsorbed is bound to be always smaller than their total number, the resultant experimental amplification factor represents a lower bound estimation of the actual enhancement factor.
O4. Assignment of the important band at 1260 cm-1 is missing in the Table 1.
We appreciate the reviewer’s observation, the assignment of this band (Ring breathing) was added to the table.
O5. The SERS spectra on Figs 3-5 seems to be baseline corrected? If it is a case, please note it in the figure captions.
Only figure 3 was corrected with Vancouver algorithm. Its figure caption was changed to:
Figure 3. (a) Raman spectrum of powder histamine after baseline correction using Vancouver algorithm, where the observed modes correspond with the work reported by Collado et al. [34]. (b) Raman spectra of histamine 1 M with (red) and without AuNPs (black).
- please explain what you mean by "analytical amplification factor 10" at conclusion?
We have revised this sentence and we concluded that this was an error from a previous version. We have dropped the “analytical amplification factor” term, and left the sentence referring only the SERS enhancement factor.
- correct some typos such as "nanostart" at conclusion.
We have fully revised the wording of the whole paper looking for misspelling and typos. We have also checked the English style in order to make the paper better readable.

Reviewer 2 Report
The paper is devoted to application of colloidal solutions of gold nanostars for SERS. Authors demonstrate theoretically and experimentally more efficient SERS signals for the nanostar systems than for systems of spherical nanoparticles. Increase of SERS response is associated with special nanoparticle shape. This is interesting result that can be considered for publication. However, authors should demonstrate more evident proves that the nanoparticle shape plays the main role in the observed effects (see the comment 4). Moreover, several positions should be clarified and improved as well.
1) Authors did not indicate dielectric properties of environment for simulation presented in Fig. 1c. Because there is a colloidal solution, the environment differs from air. It is known that spectral position of the palsmon resonance depends on surrounding matter. This has to be discussed by authors.
2) Why the value of field maximum is higher in Fig 2c than in Fig.2b? As it is followed from the caption, the fields presented in Fig.2b correspond to the resonant conditions.
3) What have colloidal concentrations been used in the experimental measurements presented in Fig. 1d?
4) It is known that increasing the size of gold spherical nanoparticles, the plasmon resonances are shifted in the red side (see for example Phys Rev. B 85, 245411 (2012)). So for the spherical nanoparticles with R around of 100 nm the plasmon resonance will be at the lambda 650 nm. What SERS signal could be expected in this case? Maybe it can provide the effect obtained for nanostars or even stronger? This has to be discussed.
Thus I consider that above comments and remarks should be clarified before a final decision is done
Author Response
Reviewer #2:
The paper is devoted to application of colloidal solutions of gold nanostars for SERS. Authors demonstrate theoretically and experimentally more efficient SERS signals for the nanostar systems than for systems of spherical nanoparticles. Increase of SERS response is associated with special nanoparticle shape. This is interesting result that can be considered for publication. However, authors should demonstrate more evident proves that the nanoparticle shape plays the main role in the observed effects (see the comment 4). Moreover, several positions should be clarified and improved as well.
O1) Authors did not indicate dielectric properties of environment for simulation presented in Fig. 1c. Because there is a colloidal solution, the environment differs from air. It is known that spectral position of the palsmon resonance depends on surrounding matter. This has to be discussed by authors.
As far as we have a colloidal solution, the surrounding medium is watar, H20. We have considered de chromatic dispersion on water’s refractive index using the Hale and Querry’s data(https://refractiveindex.info/?shelf=main&book=H2O&page=Hale).
2) Why the value of field maximum is higher in Fig 2c than in Fig.2b? As it is followed from the caption, the fields presented in Fig.2b correspond to the resonant conditions.
The resonant wavelength is at 675 nm for the nanostar and 530 for the nanosphere, while the excitation source wavelength of our Raman spectrometer is at 785 nm. We have modified the caption of Fig. 2 to make this point clearer.
3) What have colloidal concentrations been used in the experimental measurements presented in Fig. 1d?
In figure 1 d, both colloids have a concentration of 2.5 mM. This information has been added to the caption.
4) It is known that increasing the size of gold spherical nanoparticles, the plasmon resonances are shifted in the red side (see for example Phys Rev. B 85, 245411 (2012)). So for the spherical nanoparticles with R around of 100 nm the plasmon resonance will be at the lambda 650 nm. What SERS signal could be expected in this case? Maybe it can provide the effect obtained for nanostars or even stronger? This has to be discussed.
In Figure 1, the resonance response of the nanospheres alone, are modeled in the blue lines. From the figure, it is possible to observe that the resonance band is centered at around 530 nm for these gold nanospheres. Hence, not much may be expected by exciting them with the 650 nm. But We agree that a strong resonance response is expected for excitation with the 532 nm source, but as we have responded already to observation O1 of the first reviewer, the resultant accompanying fluorescence made measurements very inconvenient at this wavelength, hence, that was the reason for our effort to conduct our study exciting using the 785 nm source.
The strong electromagnetic field value showed by the nanostar is related to its structure peaks, where the electromagnetic field is focused around them. Linking this effect to plasmonic resonance of nanostar, we could increase notably the SERS signal. The spherical nanoparticles showed in Phys Rev. B 85, 245411 (2012) with an R value around of 120 nm, surrounded by air, presents is resonance at 675nm according to MIE theory. If we replace the surrounding media by water, the resonance peak moves to 600nm. Fig. A of this reply shows the scattering extinction of these different cases and the near field at these wavelengths. The maximum near field value, around 5V/m, at resonance wavelength for water medium, is notably lower than the values showed by the nanostar, around of 200V/m. Thus, considering that Raman signal goes with |E|^4, nanostars will show several orders of magnitude bigger SERS response than spherical nanoparticles.
Fig. A: a) scattering extinction for spheres surrounding by air and water. b) And c) near field values for water and air media
We have not included these plots in the manuscript because, from our point of view, it may obscure the main goal of the paper.
Thus I consider that above comments and remarks should be clarified before a final decision is done

Reviewer 3 Report
In this paper, authors report the synthesis of Au nanostars for the SERS application in the relationship of the simulations of the plasmonic optical response. Histamine molecule is chosen as the detection model showing a limit of detection (KOD) as low as 10-7 M. The manuscript is well written with solid data to support their conclusions. I am very enthusiastic to recommend the acceptance of this paper in Nanomaterials after the authors have addressed the following minor issues.
1. In page 3 and line 88, it is unclear the term [nanostars may group in clusters]. You should define the [group] and [cluster]. Electron diffraction pattern and HR-TEM image of a single Au nanostar are suggested to add this characterization.
2. Size distribution of nanostars (DLS or calculated from images) should be providing to support your suggestion of the broader bandwidth in Figure 1.d due to the particle size effect.
3. In the past research, many groups had successfully prepared star-shaped nanostructures, e.g., Nanotechnology 23 (2012) 465602; J. Am. Chem. Soc. 2015, 137, 33, 10460-10463; Journal of Colloid and Interface Science 2017, 505, 1055-1064. What are the unique structures and properties of as-prepared Au nanostar in this current work? It would be good for the authors to provide a short discussion on this, and justify the materials difference between you and other groups. This could further benefit the readers, especially to those researchers without extensive research experience on nanomaterial synthesis.
4. The EF and LOD values of this work should make relative comparison with previous works (ACS Appl. Mater. Interfaces, 2018, 10 (17), pp 14850–14856; J. Phys. Chem. C, 2008, 112 (48), pp 18849–18859; J. Phys. Chem. C, 2010, 114 (16), pp 7336–7340; J.Raman. Spectrosc.2009, 40, 86–91.
5. Besides serving as SERS detection of small molecule, gold and silver nanoparticles also have much other potential SERS applications, such as bioimaging (e.g., Sci. Rep. 2014, 4, 5593; RSC Adv., 2016, 6, 64494–64498), monitor catalysis reaction (e.g., J. Mater. Chem. A, 2018, 6, 13041–13049), and new monitor technology for bacterial activity after antibiotic treatment (Green Chem., 2018, 20, 5318–5326). It would be good for the authors to include a short discussion on the potential applications of gold and silver nanoparticles in the Introduction. This could further enrich the literature.
6. Noted that the histamine was 7.2 nmol/L at an average concentration in human plasema. This demand detection concentration is lower than the LOD using Au nanostars, only reaching to 100 nM. Is it good enough for practical application?
Author Response
In this paper, authors report the synthesis of Au nanostars for the SERS application in the relationship of the simulations of the plasmonic optical response. Histamine molecule is chosen as the detection model showing a limit of detection (KOD) as low as 10-7M. The manuscript is well written with solid data to support their conclusions. I am very enthusiastic to recommend the acceptance of this paper in Nanomaterials after the authors have addressed the following minor issues.
O1. In page 3 and line 88, it is unclear the term [nanostars may group in clusters]. You should define the [group] and [cluster]. Electron diffraction pattern and HR-TEM image of a single Au nanostar are suggested to add this characterization.
The wording of this sentence has been changed to avoid misunderstanding of the clustering effect. Now it reads as:
When prepared in colloid form, nanostars coalesce in clusters of several units (see Fig. 1.b).
2. Size distribution of nanostars (DLS or calculated from images) should be providing to support your suggestion of the broader bandwidth in Figure 1.d due to the particle size effect.
The information of the size distribution has been included in the caption of Fig. 1. The histogram is included here for clarity.
3. In the past research, many groups had successfully prepared star-shaped nanostructures, e.g., Nanotechnology 23 (2012) 465602; J. Am. Chem. Soc. 2015, 137, 33, 10460-10463; Journal of Colloid and Interface Science 2017, 505, 1055-1064. What are the unique structures and properties of as-prepared Au nanostar in this current work? It would be good for the authors to provide a short discussion on this, and justify the materials difference between you and other groups. This could further benefit the readers, especially to those researchers without extensive research experience on nanomaterial synthesis.
Taking in consideration the reviewer’s comments, we decided to implement the following change:
We have added the following sentence to the Introduction section (In line 20):
Au nanostars have been proven useful for SERS, they also present unique optical and electric properties. Previous groups have reported the synthesis of Au nanostar with a good degree of symmetry control by using a robust solution-phase method (Wenxin Niu et al, Highly Symmetric Gold Nanostars: Crystallographic Control and Surface-Enhanced Raman Scattering Property), or by increasing the seed concentration in the growth solution (Giacomo Dacarro et al. Synthesis of reduced-size gold nanostars and internalization in SH-SY5Y cells). Also, the potential of these nanostructures has been evaluated in biomedical applications by functionalizing its surface using a biocompatible polymer (Navarro et al. Synthesis of PEGylated gold nanostars and bipyramids for intracellular uptake).
4. The EF and LOD values of this work should make relative comparison with previous works (ACS Appl. Mater. Interfaces, 2018, 10 (17), pp 14850–14856; J. Phys. Chem. C, 2008, 112 (48), pp 18849–18859; J. Phys. Chem. C, 2010, 114 (16), pp 7336–7340; J.Raman. Spectrosc.2009, 40, 86–91.
We have added a couple of sentences at the introduction section to compare our results with the literature.
This result is somewhat larger than those obtained for a silver film over nanosphere (AgFON) from Wen-Chi Lin et al. [33] and far larger than that obtained from molecularly imprinted polymers by Fang Gao et al. [22] This also applies even for the values reported in refs. [J. Phys. Chem. C, 2010, 114 (16), pp 7336–7340; J. Raman Spectrosc.2009, 40, 86–91.], being our EF value closer to $1.91 \times 10^7$, recently reported [ACS Appl. Mater. Interfaces, 2018, 10 (17), pp 14850–14856].As the spectral positions …
5. Besides serving as SERS detection of small molecule, gold and silver nanoparticles also have much other potential SERS applications, such as bioimaging (e.g., Sci. Rep. 2014, 4, 5593; RSC Adv., 2016, 6, 64494–64498), monitor catalysis reaction (e.g., J. Mater. Chem. A, 2018, 6, 13041–13049), and new monitor technology for bacterial activity after antibiotic treatment (Green Chem., 2018, 20, 5318–5326). It would be good for the authors to include a short discussion on the potential applications of gold and silver nanoparticles in the Introduction. This could further enrich the literature.
We have revised the literature and included more references to refer recent advances. Our answer to query #3 also added valuable references to the bibliography of the manuscript.
6. Noted that the histamine was 7.2 nmol/L at an average concentration in human plasema. This demand detection concentration is lower than the LOD using Au nanostars, only reaching to 100 nM. Is it good enough for practical application?
In our humble opinion, the wording of the conclusion section already mention the capabilities of the proposed methods to reach physiological concentrations:
In summary, this work proposes a reliable method for the detection of histamine in concentrations as low as 10-7M, which is a concentration suitable for food quality control applications (concentrations of ≈ 10-4M). Further work and refinement of the technique are needed to detect histamine at even lower physiological concentration (≈10-9M) using the above-mentioned method.

Round 2
Reviewer 1 Report
The authors carefully addressed my point. I recommend to publish it in present form.
Reviewer 2 Report
Authors took into account all my comments.